# Validation of a SARS-CoV-2 Surrogate Virus Neutralization Test in Recovered and Vaccinated Healthcare Workers

**DOI:** 10.3390/v15020426

**Published:** 2023-02-02

**Authors:** Lina Mouna, Mehdi Razazian, Sandra Duquesne, Anne-Marie Roque-Afonso, Christelle Vauloup-Fellous

**Affiliations:** 1Virology Department, Hôpital Paul Brousse, INSERM U1193, AP-HP, Université Paris Saclay, 94804 Villejuif, France; 2Institute for Physiology and Pathophysiology, Johannes Kepler University Linz, 4040 Linz, Austria

**Keywords:** SARS-CoV-2, neutralizing antibodies, anti-spike antibody, sVNT, eCLIA, cVNT

## Abstract

Vaccination against COVID-19 is the main public health approach to fight against the pandemic. The Spike (S) glycoprotein of SARS-CoV-2 is the principal target of the neutralizing humoral response. We evaluated the analytical and clinical performances of a surrogate virus neutralization test (sVNT) compared to conventional neutralization tests (cVNTs) and anti-S eCLIA assays in recovered and/or vaccinated healthcare workers. Our results indicate that sVNTs displayed high specificity and no cross-reactivity. Both eCLIA and sVNT immunoassays were good at identifying cVNT serum dilutions ≥1:16. The optimal thresholds when identifying cVNT titers ≥1:16, were 74.5 U/mL and 49.4 IU/mL for anti-S eCLIA and sVNT, respectively. Our data show that neutralizing antibody titers (Nab) differ from one individual to another and may diminish over time. Specific assays such as sVNTs could offer a reliable complementary tool to routine anti-S serological assays.

## 1. Introduction

The severe acute respiratory syndrome coronavirus 2 (SARS-CoV-2) epidemic, which was first reported in Wuhan, China, spread rapidly worldwide and caused the COVID-19 pandemic [1]. As different vaccines become available, a clearer understanding of their efficacy in inducing sufficient public immunity against COVID-19 is of major importance [2]. The spike (S) glycoprotein of SARS-CoV-2 is the principal target of the neutralizing humoral response and is therefore used for the development of vaccines and monoclonal antibody (Ab) treatments [3,4,5,6,7]. Virus-specific neutralizing Abs (NAbs) play a key role in reducing viral replication and accelerating the viral clearance of SARS-CoV-2 [8,9]. NAbs mainly act against the receptor-binding domain (RBD) of the SARS-CoV-2 S glycoprotein, effectively blocking viral entry by preventing its binding to the angiotensin-converting enzyme 2 (ACE-2) receptor [9,10,11,12].

There is some evidence that immune protection against other human coronaviruses (HCoV) is not long-lasting, but it is also suggested that Abs against SARS-CoV-1 persist for years and display potent neutralizing activity, even in the absence of detectable IgG directed against the SARS-CoV-1 nucleocapsid protein (N) [13,14]. However, recent descriptions of SARS-CoV-2 reinfections have supported recommendations to offer vaccination to highly exposed individuals such as healthcare workers (HCW) who have recovered from a mild form of COVID-19 [15,16]. These authors also suggested that vaccination against SARS-CoV-2 will probably have a short-lived protective effect, meaning that most people should be vaccinated periodically. Until now, most serological studies have focused on hospitalized patients, but specific information on serologic responses in individuals with mild infection remains scarce and is mainly focused on seroconversion rates [17,18,19,20,21,22,23,24]. Marot et al. recently showed that the neutralizing activity of Ab appears to be transient with a decline, or even loss of NAb titers from two months after disease onset, thus supporting the recommendation to vaccinate infected HCW [25]. However, in the context of vaccine shortages and/or a mistrust of vaccines, it might be highly beneficial to determine who really needs a boost and how many doses might be necessary depending on their clinical history together with accurate serological information. Serological testing, especially to detect NAbs, is therefore essential to identify individuals who are potentially at a lower risk of hospitalization or serious COVID-19 infection.

In COVID-19 patients, NAbs can be detected within two weeks of symptom onset, but in some cases this is substantially longer [26,27,28,29]. However, the dynamics of NAbs and their correlation with anti-S Ab have not been explored in COVID-19 patients more than six months after the onset of symptoms. Moreover, it is still unclear whether current serological assays that detect anti-S Ab will predict neutralizing activities or protection against re-infection with the virus [30].

NAbs can be detected using conventional virus neutralization tests (cVNTs), and their presence is often correlated with protective immunity [31]. cVNTs are considered to be the gold standard [21] but they must be performed in a biosafety level-3 laboratory (BSL-3); they are time-consuming, labor-intensive, require several days of work by highly skilled operators and are hardly standardized when compared to other serologic assays. Very few laboratories can therefore run such tests, which is not appropriate for mass use. On the other hand, enzyme-linked immunosorbent assays (ELISA), and ELISA variants such as the lateral flow assay (LFA) and chemiluminescence immunoassay (CLIA) can detect anti-S or anti-RBD Ab with high sensitivity but vary in their ability to predict NAb activity [32,33,34,35,36]; thus, the results may not directly correlate with protection [37,38].

During the present study we evaluated the analytical and clinical performance of a surrogate virus neutralization test (sVNT) (iFlash-2019-nCoV NAb assay, YHLO, China) that had been designed to detect total NAbs in an isotype and species-independent manner and can be completed in 1–2 h in a BSL-2 laboratory [39]. The anti-S eCLIA assay (Elecsys anti-SARS-CoV-2 S, Roche Diagnostics, Switzerland) can detect total binding antibodies but not specifically NAb. Indeed, although RBD might be the target for NAb, some RBD Ab cannot fully block the interaction between RBD and ACE2, which might interfere with re-infection prevention. We compared cVNT, sVNT and anti-S eCLIA assays in immunocompetent subjects (mainly healthcare workers) at 9–11 months after a COVID-19 infection, after the first dose of vaccine, after the second dose of vaccine and in patients with a history of COVID-19 after one dose of vaccine.

## 2. Materials and Methods

This study was conducted in the Virology Laboratory at Paul Brousse Hospital (Paris, France) in accordance with the requirements of the Declaration of Helsinki as a retrospective non-interventional study with no addition to standard-of-care procedures. The reclassification of biological remnants into research material after completion of the virological tests that had been ordered was registered under number DC 2009-965 and received ethical approval from CPP Ile de France 7 (N°CO-15-000) in accordance with French law.

### 2.1. Clinical Samples

First of all, 54 sera were selected to determine three thresholds for anti-S eCLIA and sVNT values predictive of neutralizing activity, using a cVNT as the gold standard (Figure 1):11 serum samples from vaccinated individuals, collected at least 14 days after the first dose (n = 7) or at least 7 days after the second dose (n = 4); BNT162b2 mRNA COVID-19 vaccine;41 serum samples from recovered COVID-19 patients collected at least 3 months after the onset of infection (n = 36) or at least 6 months after the onset of infection and 7 days after the first vaccine dose (n = 5); BNT162b2 mRNA COVID-19 vaccine;As a control, two samples were collected from unaffected and non-vaccinated individuals.

In a second step, 394 samples were collected to compare the results of the anti-S eCLIA and sVNT assays (Figure 2):103 serum samples collected before the COVID-19 pandemic (2019) including three serum samples from patients with other HCoV infections (2 HcoV-OC43, 1 HcoV-NL63);27 serum samples collected at least three weeks after infection from patients with a confirmed SARS-CoV-2 variant (alpha, beta, or delta);168 serum samples collected from infection-naïve HCW before vaccination (n = 73), at least 14 days after the first dose of vaccine (n = 36) and at least seven days after the second dose of vaccine (n = 59);96 serum samples from COVID-19 recovered HCW collected 9–11 months after the onset of infection (n = 73) and at least seven days after one dose of vaccine (n = 23).

The median age of the HCW was 53 years (range: 22 to 86 years) and 64% were women. None were immunocompromised. All were vaccinated with the BNT162b2 mRNA COVID-19 vaccine.

### 2.2. Conventional Virus Neutralization Test (cVNT)

All procedures related to cVNTs were performed in a BSL-3 laboratory in accordance with WHO guidelines. Vero E6 cells (African green monkeys, kidney) were grown at 37 °C under 5% CO_2_ in Dulbecco’s modified eagle medium (DMEM, 319966), supplemented with L-Glutamine, 10% fetal bovine serum (FBS, FCS, Sigma-Aldrich, #F9665), and 1% penicillin/streptomycin/neomycin (PSN, GIBCO 15640055). The SARS-CoV-2 virus was isolated from a RT-PCR-confirmed COVID-19 patient hospitalized in Paul Brousse Hospital (Villejuif, France) in December 2020, and amplified in Vero cells. We confirmed the virus titer using TCID50 (10 7.5 PFU/mL) and RT PCR (CT: 9.5). Serum samples were decomplemented by heat inactivation (56 °C for 30 min) and subjected to serial two-fold dilution in PBS 1X (1:2 to 1:2048, each dilution 250 µL). Different serum dilutions were incubated with 250 µL diluted virus (103 TCID50/mL) in DMEM with 2% FBS at 37 °C, under 5% CO_2_ for 1 h, and then 100 µL of each dilution was added to Vero cells seeded (3 × 104/well) in 96-well plates (four technical replicates for each sera dilution; n:4). The plates were incubated at 37 °C under 5% CO_2_ for 96 h until microscopic examination on day 4 to assess the cytopathic effect (CPE). The NAbs titer was confirmed with the highest dilution of sera that inhibited 100% of the CPE (absence of cytopathic effect). cVNT titers ≥1:16 were considered to be positive for SARS-CoV-2 NAbs.

### 2.3. Surrogate Virus Neutralization Test (sVNT)

The iFlash-2019-nCoV NAb assay is a paramagnetic particle chemiluminescent one-step immunoassay for the quantitative determination of NAb in serum using an automated analyzer. This assay is based on the Ab-mediated blockage of virus–host interactions between the ACE2 receptor protein and the RBD of the viral S protein. It is designed to mimic the virus–host interaction by direct protein–protein interactions in a test tube. In brief, NAb in the sample reacts with RBD antigen-coated microparticles to form a complex. In a second step, an ACE2 conjugate is added to competitively bind to the RBD-coated particles which have not been neutralized by NAb from the sample. Bound particles are then detected through a chemiluminescent reaction, and a calibration curve enables quantification of the amount of NAb. Results are expressed in IU/mL with the manufacturer providing a cut-off point of 24 IU/mL.

### 2.4. Elecsys Anti-S eCLIA Assay (Anti-S eCLIA)

The anti-S eCLIA assay was used for the quantitative determination of antibodies to the SARS-CoV-2 S protein receptor-binding domain (RBD). The quantification range of anti-S eCLIA is 0.4 to 250 U/mL (results < 0.4 U/mL are considered to be non-reactive, and anti-S titers outside the quantification range are expressed as >250 U/mL). The manufacturer’s positive cut-off point is 0.4 U/mL.

### 2.5. Statistical Analysis

Statistical analysis was performed with GraphPad Prism Software 9.1.2 (GraphPad, La Jolla, CA, USA). The Wilson-Brown statistical method with a 95% confidence interval (*p* value < 0.0001) was performed. A correlation test (Spearman) and receiver operating characteristic curve (ROC) were conducted using Analyse-it software v5.65. Results with *p* values < 0.05 were considered to be significant.

## 3. Results

### 3.1. Correlation of Anti-S eCLIA, sVNT and cVNT Results

Fifty-four samples from naïve, infected, and/or vaccinated patients were used to assess the correlation between anti-S eCLIA, sVNT and cVNT (Figure 1). Overall, 3/54 had no detectable anti-S eCLIA (titer <0.4 U); the remaining samples had titers ranging from 1.21 to > 250 U/mL. sVNTs were performed on 45/54 samples with titers ranging from 0 to 12,377.784 IU/mL. The control samples displayed no neutralizing activity and no reactivity in any immunoassay. cVNTs were performed on these 54 samples with neutralization results that ranged from 0 to 1:1024, and 20/54 (37%) achieved significant neutralization in a dilution ≥1:16.

The anti-S eCLIA and sVNT titers were correlated (Spearman’s rs = 0.761, *p* < 0.0001). Both the Roche and iFlash immunoassays were able to identify cVNT serum dilutions >1:16, with AUC (area under the curve) values of 0.950 (95% CI 0.889–1.011), and 0.870 (95% CI 0.752–0.988) (*p* < 0.0001) for anti-S eCLIA and sVNTs, respectively. The immunoassays were equally discriminant. The optimal thresholds to identify cVNT titers ≥ 1:16 were 74.5 U/mL and 49.4 IU/mL for anti-S eCLIA and sVNTs, respectively (so-called Nabs cut-offs). Applying these thresholds, sensitivity and specificity reached 95% and 93.4% for anti-S eCLIA, and 82.4 and 85.7% for sVNTs (Figure 3).

### 3.2. Performance of sVNT vs. cVNT

The specificity and sensitivity of sVNTs were assessed on sera collected from patients before the COVID-19 pandemic (expected to be negative) and from patients who had recovered following a confirmed COVID-19 infection (expected to be positive). All those expected to be negative were indeed negative (103/103; 100% specificity), including the samples collected from patients with other HCoV infections. Among the 100 sera taken from recovered patients, 91% had detectable titers, including 64/73 (87.7%) samples collected ≥ 9 months after infection, and 27/27 (100%) samples from patients with recent confirmed COVID-19 infection by a variant.

### 3.3. Anti-S eCLIA vs. sVNT

Seventy-three serum samples were collected before vaccination. All (73/73) were negative in the Roche and sVNT assays with mean titers <0.4 U/mL and 10.82 ± 5 IU/mL, respectively (Table 1).

At least 14 days after the first vaccine dose and before the second dose, 28/36 (78%) of the serum samples displayed positive anti-S eCLIA titers (ranging from 2.88 to >250 U/mL; mean titer: 64.78 ± 90), and 21/36 (58%) were positive with sVNTs (titers ranging from 24.792 to 1188.864 IU/mL; mean titer: 86.26 ± 198). Based on established NAb cut-offs, the number of individuals with potential protective antibody levels fell to 12/36 (33%) with sVNTs and to 10/36 (28%) with anti-S eCLIA. At least seven days after the second dose, 55/59 (93%) displayed positive anti-S eCLIA titers (ranging from 1 to >250 U/Ml; mean titer: 201.58 ± 98)), and 51/59 (86%) were positive with sVNTs (titers ranging from 26.88 to 240,000,000 IU/mL; mean titer: 36,611,419 ± 82,643,261)). Based on established NAb cut-offs, 50/51 (85%) individuals had sVNT titers >50 IU/mL and 47/59 (80%) had an anti-S eCLIA >75 U/mL.

Between 9 and 11 months after recovering from COVID-19 infection, 73 samples were collected from COVID-19 healthcare workers. All were symptomatic with mild symptoms (mainly cough, fever, breathlessness, aching body, headache, diarrhea or anosmia). Overall, 72/73 (99%) displayed the presence of anti-S eCLIA (titers ranging from 10.39 to > 250 U/mL; mean titer: 169 ± 92), and 64/73 (88%) were positive with sVNTs (titers ranging from 25 to 2697 IU/mL; mean titer: 394 ± 755). Based on established NAb cut-offs, the number of individuals with protective antibody levels fell to 42/73 (58%) with sVNTs and to 56/42 (77%) with anti-S eCLIA. Among these recovered COVID-19 HCW, 23 had received one dose of vaccine at least 6 months after the onset of infection. All (23/23; 100%) displayed the presence of anti-S eCLIA (titers ranging from 60.89 to > 250 IU/mL; mean titers: 250 ± 41) and positive sVNTs (titers ranging from 35.712 to 240,000,002.4 IU/mL; mean titer: 31,306,349 ± 85,874,735). Based on established NAb cut-offs, 22/23 (96%) displayed sVNT titers >50 IU/mL and anti-S eCLIA titers >75 U/mL.

In conclusion, anti-S eCLIA Abs levels from hybrid immunized individuals (infected–vaccinated) and two-dose vaccine immunized individuals were higher than those found in other situations (*p* < 0.03). NAb levels (sVNT) in sera from hybrid immunized individuals (infected–vaccinated), two-dose vaccine immunized individuals and patients who had recovered from variants of SARS-CoV-2 infection were also higher than in the other situations. (*p* < 0.0003) (Figure 4; Table 1).

## 4. Discussion

We evaluated analytical and clinical performance of a surrogate virus neutralization test (sVNT) by comparison with a conventional virus neutralization test (cVNT), before studying the neutralizing activity of the serum samples, a biological marker that is often correlated to protective immunity [31]. Both assays can determine the functional ability of Ab to prevent viral binding to the ACE2 receptor. These assays also have the advantage that they can study overall serum neutralizing activity (both isotypes and epitope-specific Ab). However, cVNTs cannot be performed routinely on a large number of samples, unlike the automated sVNT assay. Our results indicated that sVNTs displayed high specificity and no cross-reactivity. We also established that a threshold of 50 IU/mL with sVNTs was predictive of the neutralizing activity of anti-SARS-CoV-2 Ab in cell culture (vs. the 24 IU/mL cutoff point recommended by the manufacturer) with sensitivity and specificity reaching 82.4% and 85.7%, respectively.

It is well known that the positive results of current large-scale serological ELISA or CLIA assays, even those which specifically target the S protein of SARS-CoV-2, do not necessarily indicate neutralizing immunity because they detect some but not all NAb and may also detect non-NAb. Indeed, although RBD might be the target for NAb, some RBD Ab cannot fully block the interaction between RBD and ACE2, which might fail in preventing re-infection [40]. With the anti-S eCLIA assay, we were able to establish a threshold of 75 IU/mL anti-S, which was related to the neutralizing activity of anti-SARS-CoV-2 antibodies in cVNTs with sensitivity and specificity of 95% and 93.4%, respectively. This threshold differs quite markedly from the 0.8 IU/mL cut-off point recommended by the manufacturer and will certainly have an impact on the clinical interpretation of results.

Our understanding of the duration and nature of protective immunity against SARS-CoV-2 is currently very limited. The kinetics of antibody-mediated immunity to SARS-CoV-2 infection, and how long this may last, remain unknown. Working in ferrets, some authors recently showed that re-infected SARS-CoV-2 animals with high Nabs displayed attenuated viral replication and rapid viral clearance. Direct-contact transmission was only observed from re-infected ferrets with low NAb titers, demonstrating a close correlation between NAb titers and protection in an animal model [40]. Although a recent study in a rhesus macaque model did not find any evidence of reinfection and therefore suggested a robust protective immune response when the animals were re-exposed to SARS-CoV-2 one month after the initial viral infection [41], only monkeys with high NAb titers were used for the reinfection studies and a few of them displayed low levels of replication following reinfection. Overall, these studies indicate that the NAb titer is a critical determinant in providing protection against reinfection; initial exposure to other human coronaviruses are known to fail in eliciting a sufficient protective immune response, in some cases with a higher viral load [42]. This result, consistent with our findings and those of other groups, indicates that SARS-CoV-2 infection elicits a neutralizing activity of sera, at least temporarily, which may be correlated with a protective humoral response.

Our data showed that NAb titers in patients may vary and diminish over time depending on different individuals, which is in accordance with findings in patients infected with other human HCoV [43,44]. The short-term humoral immune response in COVID-19 patients is also highly consistent with that observed in patients infected with SARS-CoV-1 and MERS-CoV [45,46] who showed a rapid decrease in virus-specific antibody titers within 3–4 months. Among the 73 recovered patients in our study, 64 (88%) displayed low NAb titers (mean titer: 394 ± 755) 9 to 11 months after infection, suggesting that the Ab titers may diminish over time, or that some recovered patients may not produce a high-titer neutralizing response during SARS-CoV-2 infection. Although a positive correlation between NAb titers and IgG has been shown in patients within three months of infection, our results suggest that more than nine months after infection, anti-S serological tests may fail to find NAb and that specific assays such as a sVNT may offer a valuable complementary tool [47]. However, their response after one booster vaccine (mean titer: 31,306,349.36 ± 85,874,735) revealed equivalent NAb titers compared to naïve patients who received two doses of vaccine (mean titer: 36,611,419.09 ± 82,643,261). This finding encourages the performance of serological tests in the general population in order to identify individuals who might have experienced asymptomatic infection in order to optimize vaccination schedules, as is now recommended in most countries.

As the COVID-19 pandemic continues to spread worldwide, most people infected with SARS-CoV-2 will recover from primary infection. To hamper the continuous spread of this virus, recovered individuals may have sufficient NAb to protect themselves against reinfection. Recent studies have reported that asymptomatic COVID-19 patients exhibit a weaker Ab response than patients with severe disease. Moreover, a rapid decline of Ab in asymptomatic patients has been seen when compared to severe COVID-19 patients [48,49,50]. sVNTs may therefore be more reliable for the assessment of herd immunity and humoral protective immunity in recovered/vaccinated patients. As sVNTs will not be available in all laboratories, its use could be considered for the routine revaluation of serological assays in order to facilitate screening in the general population and target individuals who require one or two doses of vaccine. Further, in patients who are known to have a weak immune response to vaccination (e.g., those undergoing dialysis or receiving rituximab), it would be helpful to identify individuals in whom a third dose of vaccine is warranted.

Although cases of suspected SARS-CoV-2 reinfection have continued to rise among recovered COVID-19 patients, their immune response to the virus, and particularly the role of Nab, has not been fully characterized [51]. sVNTs have been reported recently and they are already commercially available, but its value needs to be analyzed and assessed for each specific context [52]. Our understanding of the duration and nature of protective immunity to SARS is currently very limited. Our data suggest that NAb may be absent in some patients who have recovered from COVID-19 and in these specific individuals, a single booster vaccination will enhance their pre-existing immunity and reach a protective level. Given the current situation where a significant proportion of the population continues to display a degree of mistrust in vaccines, or in the event of vaccine shortages, sVNT assays or routine serological assays with newly established NAb cut-offs would avoid the vaccination of individuals whose titers are indicative of protective immunity. Studies are currently ongoing to evaluate the usefulness of sVNTs in immune patients at risk of severe COVID-19.

The present study had several limitations, such as a lack of systematic follow-up and limited sequential sampling. More studies are necessary to assess the protection correlates after SARS-CoV-2 infection, such as the minimal NAb titer required for protection. Further, it will be pertinent to study the protective role of NAb during reinfection in recovered COVID-19 patients and the role of the cellular immune response. It will also be important to assess changes in the levels of antibodies directed against the S protein and RBD over time, in order to improve our understanding of the potential protective immune response to SARS-CoV-2 infection.

## Figures and Tables

**Figure 1 viruses-15-00426-f001:**
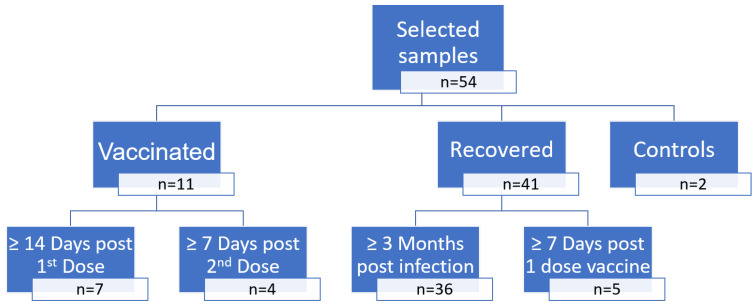
Samples selected to determine thresholds of anti-S eCLIA and sVNT values predictive of neutralizing activity using a cVNT as a gold standard.

**Figure 2 viruses-15-00426-f002:**
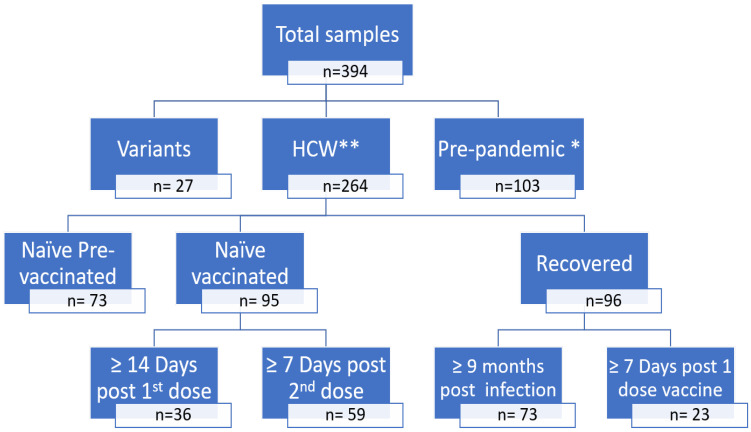
Samples selected to compare anti-S eCLIA and sVNT results. * Including 3 other HcoV, ** HCW—healthcare workers.

**Figure 3 viruses-15-00426-f003:**
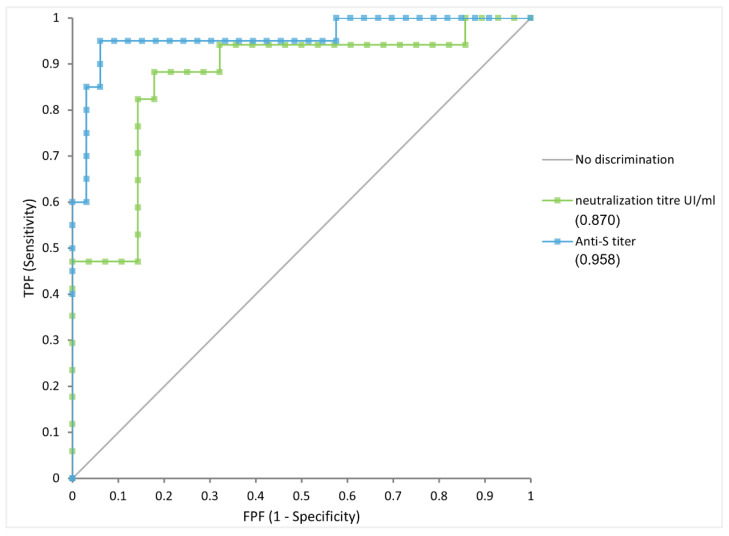
Receiver operating characteristic curve (ROC) for prediction of neutralization titers ≥1:16 by cVNT based on the anti-S eCLIA and sVNT, AUC (area under the curve); anti-S eCLIA: 0.958 (95% CI 0.889–1.011); sVNT: 0.870 (95% CI 0.752–0.988) (*p* < 0.0001).

**Figure 4 viruses-15-00426-f004:**
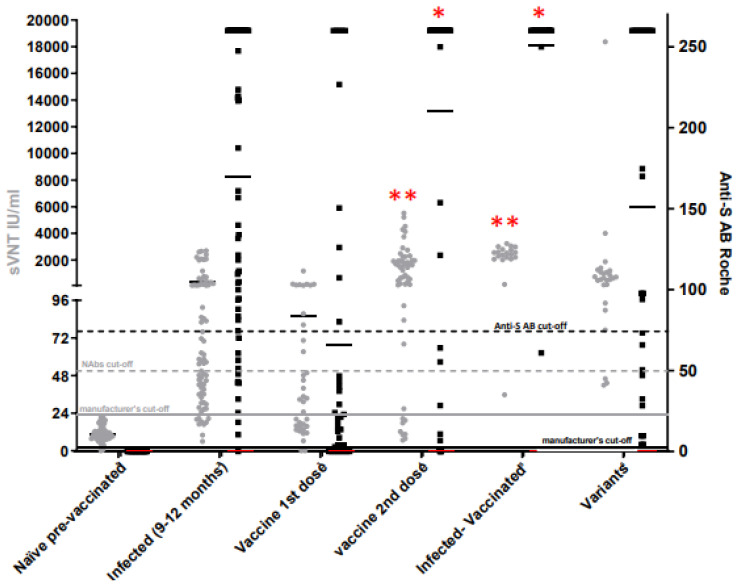
Anti-S eCLIA (black dots) vs. sVNT (grey dots) results. Cut-offs provided by manufacturers and cut-offs established with cVNTs are represented in full and dotted lines, respectively. Means of titers for each population and each assay are represented by a dash (**p* < 0.05; ** *p* < 0.01).

**Table 1 viruses-15-00426-t001:** Comparison between sVNT and anti-S eCLIA titers in different HCW populations.

	sVNT			Anti-S Abs		
	Mean Titer (Positive Cut-Off: >24 IU/mL)	N° and % (>24 IU/mL)	N° and % (>50 IU/mL)	Mean titer (Positive Cut-Off: >0.8 U/mL)	N° and % (>0.8 U/mL)	N° and % (>75 U/mL)
Naïve pre-vaccination	10.82 ± 5	0 (0%)	0 (0%)	<0.4	0 (0%)	0 (0%)
14 days post 1st Dose	86.26 ± 198	21 (58%)	12 (33%)	64.78 ± 90	28 (78%)	10 (28%)
7 days post 2nd Dose	36,611,419.09 ± 82,643,261	51 (86%)	50 (85%)	201.58 ± 98	55 (93%)	47 (80%)
Infected 9–12 months	394 ± 755	64 (88%)	42 (58%)	169 ± 92	72 (99%)	56 (77%)
Infected 7 days post 1st Dose	31,306,349.36 ± 85,874,735	23 (100%)	22 (96%)	250 ± 41	23 (100%)	22 (96%)

## Data Availability

Data can be provided by the authors on demand.

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
