# Peer review of "Validation of a SARS-CoV-2 Surrogate Virus Neutralization Test in Recovered and Vaccinated Healthcare Workers"

_viruses, 2023, doi:10.3390/v15020426_

Round 1
Reviewer 1 Report
In this study, the author evaluated the antibody response in SARS-CoV-2 recovered and vaccinated cohort, by using sVNT, compared to cVNT and eCLIA. One interesting finding is that they identified the optimal thresholds for sVNT and eCLIA, mapping to cVNT titer. However, there are several concerns that need to be addressed.
Major points:
1. In line 74-76, some SARS-CoV-2 antibodies, like class 3 antibody, binds outside the ACE2-binding site, but still has high neutralizing ability (PMID: 33045718).
2. There is no evidence in this study, or others, showing which level of the neutralizing antibody has a protective effect in humans, then the author can not use the term "protective antibody level" in the manuscript, like in line 202-204.
3. In line, 240-241, the author doesn't show evidence for the cross-reactivity.
4. The topic of this study is the comparison among cVNT, sVNT and eCLIA, then the author may need to add the correlation between every two methods.
5. In the discussion, the author may discuss the previously published paper about the comparison between sVNT and the other method, like PMID: 33043818.
Minor points:
1. In line 135-136, any reference for the positive cutoff?
2. In Figure 4, the author needs to do the statistical analysis.
3. In line 303-307, there is "SARS-CoV-2", but not "SARS".
Author Response
We would like to thank the reviewers for their valuable comments and suggestions for improvements to our text. We have addressed their responses comments individually, and revised the article to incorporate their suggestions and to make the necessary clarifications.
Author's Reply to the Review Report (Reviewer 1)
Comments and Suggestions for Authors
In this study, the author evaluated the antibody response in SARS-CoV-2 recovered and vaccinated cohort, by using sVNT, compared to cVNT and eCLIA. One interesting finding is that they identified the optimal thresholds for sVNT and eCLIA, mapping to cVNT titer. However, there are several concerns that need to be addressed.
Major points:
- In line 74-76, some SARS-CoV-2 antibodies, like class 3 antibody, binds outside the ACE2-binding site, but still has high neutralizing ability (PMID: 33045718).
Response: the sentence has been modified accordingly
- There is no evidence in this study, or others, showing which level of the neutralizing antibody has a protective effect in humans, then the author can not use the term "protective antibody level" in the manuscript, like in line 202-204.
Response: The word ʺpotentialʺ has been added to the phrase: protective antibody levels
- In line, 240-241, the author doesn't show evidence for the cross-reactivity.
Response: The specificity and sensitivity of sVNT were assessed on sera collected from patients before the COVID-19 pandemic (expected to be negative) and from patients who had recovered following a confirmed COVID-19 infection (expected to be positive). All those expected to be negative were indeed negative (103/103; 100% specificity), including the samples collected from patients with other HCoV infections (lines 104-105 and 187).
- The topic of this study is the comparison among cVNT, sVNT and eCLIA, then the author may need to add the correlation between every two methods.
Response: 54 Samples were selected to determine thresholds of anti-S eCLIA and sVNT values predictive of neutralizing activity using cVNT as a gold standard. Correlation of anti-S eCLIA, sVNT and cVNT results was conducted in line 163-178
- In the discussion, the author may discuss the previously published paper about the comparison between sVNT and the other method, like PMID: 33043818.
Response: We have modified the conclusion accordingly.
ʺNeutralization tests using recombinant RBD acting as a surrogate of real virus have recently been reported and they are already commercially available but its value needs to be analyed and assessed for each specific contextʺ.
Minor points:
- In line 135-136, any reference for the positive cutoff?
Response: Using Receiver operating characteristic curve (ROC), prediction of neutralization titers ≥1:16 by cVNT was based on the anti-S eCLIA and sVNT, AUC (Area under the curve); anti-S eCLIA : 0.958 (95% CI 0.889-1.011); sVNT : 0.870 (95% CI 0.752-0.988) (p<0.0001).
- In Figure 4, the author needs to do the statistical analysis.
Response: Modification has been done
- In line 303-307, there is "SARS-CoV-2", but not "SARS".
Response: Modification has been made

Reviewer 2 Report
The authors evaluated clinical performance of sVNT compared to cVNT and anti-s-eCLIA, focusing on correlation and AUC data. I have several concerns regarding the methodology and results.
1. For sVNT, a cut-off value of 24 IU/mL was provided in the Methods section. Authors should present clinical performance (sensitivity and specificity) using this cutoff to predict cVNT outcome.
2. 2.4 Elecsys anti-S eCLIA assay (lines 149-154); Did the author retest after dilution in samples > 250 U/mL?
3. 3.1 Resul; Authors should provide correlation plots comparing cVNT with sVNT and CLIA.
4. 3.2 -3.3 results and conclusion. Authors should present the sensitivity and specificity of sVNT and eCLIA to predict cVNT according to the appropriate cut-off values
Author Response
We would like to thank the reviewers for their valuable comments and suggestions for improvements to our text. We have addressed their responses comments individually, and revised the article to incorporate their suggestions and to make the necessary clarifications.
Author's Reply to the Review Report (Reviewer 2)
Comments and Suggestions for Authors
The authors evaluated clinical performance of sVNT compared to cVNT and anti-s-eCLIA, focusing on correlation and AUC data. I have several concerns regarding the methodology and results.
- For sVNT, a cut-off value of 24 IU/mL was provided in the Methods section. Authors should present clinical performance (sensitivity and specificity) using this cutoff to predict cVNT outcome.
Response: cVNT are considered as the gold standard. Therefore, first of all, a correlation study of anti-S eCLIA, sVNT was conducted based on cVNT results (3.1 results). Established NAb cut-off (>50 IU/ml ) were based on cVNT positive titers ≥1:16. We didn’t take in concideration sVNT clinical performance with a cut-off of 24 IU/Ml because of lack of cVNT positive titers.
- 4 Elecsys anti-S eCLIA assay (lines 149-154); Did the author retest after dilution in samples > 250 U/mL?
Response: No, we did not make any dilution.
- 3.1 Resul; Authors should provide correlation plots comparing cVNT with sVNT and CLIA.
Response: 54 Samples were selected to determine thresholds of anti-S eCLIA and sVNT values predictive of neutralizing activity using cVNT as a gold standard. Correlation of anti-S eCLIA, sVNT and cVNT results was conducted in line 163-178
- 2 -3.3 results and conclusion. Authors should present the sensitivity and specificity of sVNT and eCLIA to predict cVNT according to the appropriate cut-off values
Response: cVNT are considered as the gold standard. Therefore, first of all, a correlation study of anti-S eCLIA, sVNT was conducted based on cVNT results. Established NAb cut-off (75 IU/ml &>50 IU/ml ) were based on cVNT positive titers ≥1:16 (3.1 results).

Round 2
Reviewer 1 Report
The author addressed all of my concerns.
Author Response

(The authors gave the same response as above.)

Reviewer 2 Report
The authors did not fully respond to my concerns and provided no further clarification to make this manuscript suitable for publication.
Author Response
We would like to thank the reviewers for their valuable comments and suggestions for improvements to our text. We have addressed their responses comments individually, and revised the article to incorporate their suggestions and to make the necessary clarifications.
Author's Reply to the Review Report (Reviewer 2)
Comments and Suggestions for Authors
The authors evaluated clinical performance of sVNT compared to cVNT and anti-s-eCLIA, focusing on correlation and AUC data. I have several concerns regarding the methodology and results.
Second request: 2.4 Elecsys anti-S eCLIA assay (lines 149-154); Did the author retest after dilution in samples > 250 U/mL?
Response: No, we did not make any dilution
First, third and fourth requests:
For sVNT, a cut-off value of 24 IU/mL was provided in the Methods section. Authors should present clinical performance (sensitivity and specificity) using this cutoff to predict cVNT outcome.
3.3.1 Results; Authors should provide correlation plots comparing cVNT with sVNT and CLIA.
3.2 -3.3 results and conclusion. Authors should present the sensitivity and specificity of sVNT and eCLIA to predict cVNT according to the appropriate cut-off values
Response: We firstly caracterized a panel of 54 serum with cVNT which is always considered as the gold standard. Indeed the cut-off value provided by manufacturer for sVNT is 24 IU/ml. However, after correlation between sVNT and cVNT (section “3.1 results”) we demonstrated that a cut-off of 50 IU/ml is better correlated to immune status based on cVNT positive titers ≥1:16. Finally, correlation of anti-S eCLIA and sVNT versus cVNT was conducted (cf lines 163-178).
